# Elevated binding and functional antibody responses to SARS-CoV-2 in infants versus mothers

Caitlin I. Stoddard [1], Kevin Sung[2], Zak A. Yaffe[1,3], Haidyn Weight[1], Guillaume Beaudoin-Bussières[4,5], Jared Galloway [2], Soren Gantt[5,6], Judith Adhiambo[7], Emily R. Begnel[8], Ednah Ojee[7], Jennifer Slyker[8], Dalton Wamalwa[7], John Kinuthia[8,9], Andrés Finzi[4,5], Frederick A. Matsen IV [2,10], Dara A. Lehman[1,8,11] ✉ & Julie Overbaugh [1,2,11] ✉

Infant antibody responses to viral infection can differ from those in adults. However, data on the specificity and function of severe acute respiratory syndrome coronavirus 2 (SARS-CoV-2) antibodies in infants, and direct comparisons between infants and adults are limited. Here, we characterize antibody binding and functionality against Wuhan-Hu-1 (B lineage) strain SARS-CoV-2 in convalescent plasma from 36 postpartum women and 14 of their infants infected with SARS-CoV-2 from a vaccine-naïve prospective cohort in Nairobi, Kenya. We find significantly higher antibody titers against SARS-CoV-2 Spike, receptor binding domain and N-terminal domain, and Spike-expressing cell-surface staining levels in infants versus mothers. Plasma antibodies from mothers and infants bind to similar regions of the Spike S2 subunit, including the fusion peptide (FP) and stem helix-heptad repeat 2. However, infants display higher antibody levels and more consistent antibody escape pathways in the FP region compared to mothers. Finally, infants have significantly higher levels of antibody-dependent cellular cytotoxicity (ADCC), though, surprisingly, Spike pseudovirus neutralization titers between infants and mothers are similar. These results suggest infants develop distinct SARS-CoV-2 binding and functional antibody activities and reveal age-related differences in humoral immunity to SARS-CoV-2 infection that could be relevant to protection and COVID-19 disease outcomes.

Antibody responses to viral infection often differ between infants and adults[1–5], owing to several factors, including the developing infant immune system and differences in infection exposure history. Relatively little is known about infant-specific antibody responses to SARS-CoV-2, which could contribute to the age-dependent severity of coronavirus disease 2019 (COVID-19)[6–8]. Plasma antibodies from individuals infected with SARS-CoV-2 target several viral proteins, though antibodies targeting the surface glycoprotein, Spike, are likely correlates of protection based on vaccine and SARS-CoV-2 challenge studies (reviewed in[9]). Thus, characterizing the levels and functional capacity of antibodies that bind to Spike, and its subdomains, is important for understanding humoral immunity to SARS-CoV-2 across the age spectrum.

Several common antibody binding sites have been identified within the two subunits of Spike (S1 and S2). These include epitopes within the receptor binding domain (RBD) of S1, and more conserved

regions of the S2 subunit, including the SARS-CoV-2 fusion machinery, which appears to be less subject to mutation[10–12]. While most neutralizing antibodies against SARS-CoV-2 target the RBD, the majority of plasma antibodies bind elsewhere on Spike[13–15]. Antibodies targeting sites outside of the RBD, including those in the S2 subunit with documented neutralizing activity or Fc-mediated effector functionality[16–18], are attractive therapeutic candidates because there has been no evidence of escape as SARS-CoV-2 continues to evolve. Several studies have identified the fusion peptide (FP), heptad repeats 1 and 2 (HR1 and HR2), and the stem helix (SH-H), which partially overlaps with the N-terminus of HR2, as targets of S2-directed antibody responses in adults[13,19,20]. Whether these are also prominent antibody targets in infants, and whether infants and their mothers differ in antibody binding profiles at the epitope level has not been examined.

While prior studies have assessed neutralization capacity in cohorts that include older children[21–24], few studies have assessed SARS-CoV-2 antibody function[25] in infants early in life or directly compared antibody responses in infants and adults. Though antibody neutralization of SARS-CoV-2 remains a key component of protective and therapeutic immunity, there is increasing evidence for the importance of non-neutralizing antibody effector functions, such as Fc receptor-mediated antibody-dependent cellular cytotoxicity (ADCC) in protection against SARS-CoV-2[6,26–30]. This is true for other viral infections as well; HIV-specific ADCC activity in multiple studies has been associated with improved outcomes in infants living with HIV[31–34]. Thus, there is a need for detailed characterization of age-related commonalities and differences in both binding and functional antibody responses to SARS-CoV-2 infection.

In this work, we examine the properties of the antibody response to SARS-CoV-2 infection in infants versus their mothers within a single cohort study. We show that infants have distinct antibody responses compared to mothers, including elevated levels of antibody binding to Spike, elevated non-neutralizing antibody activity (ADCC), and convergent antibody binding escape profiles in the FP region of the Spike protein.

## Results

### Participant groups and seropositive sample identification

Longitudinal plasma samples collected from infants and their mothers enrolled in a Nairobi, Kenya-based prospective cohort study (the Linda Kizazi study) were tested previously for SARS-CoV-2 seropositivity by nucleocapsid enzyme-linked immunosorbent assay (ELISA)[35]. The first seropositive sample from individuals who seroconverted during the original study period was included in this study (Table 1). Sample collection occurred at roughly three-month intervals, and we estimated the time of infection by calculating the midpoint between the last seronegative sample and the first seropositive sample for each individual[35]. Because antibody responses can wane significantly over time[36,37], we compared time since infection at the time of sampling included in this study and found no significant difference between estimated days since in infection in infants versus mothers, suggesting the degree of antibody waning was similar between the two groups (Fig. S1).

Importantly, all mothers who seroconverted to SARS-CoV-2 did so after giving birth, and thus any antibodies detected in the infant were not due to passive transfer from their mother. Likewise, antibodies present in human breast milk do not circulate systemically in infants in appreciable amounts[38]. Mothers in the cohort were either living with HIV and on antiretroviral therapy (ART) for ≥6 months prior to enrollment or not living with HIV, and infants were HIV-exposed/uninfected or unexposed (Table 1). HIV status in this cohort was not found to influence the risk of SARS-CoV-2 infection, no participants were vaccinated against SARS-CoV-2, and whole genome sequencing from stool revealed the B.1 lineage to be present in two of the samples, consistent with global circulation patterns at the time of sample collection[35]. All cases of COVID-19 were either asymptomatic or mild in disease severity (Table 1).

### SARS-CoV-2 antibody binding in seropositive infants and mothers

To compare SARS-CoV-2 antibody titers between infants and mothers, we tested their first seropositive plasma samples via two methods: (1) a commercially available multiplexed electrochemiluminescence platform (MSD) to detect IgG binding to SARS-CoV-2 antigens including full-length Spike, RBD, N-terminal domain (NTD), and nucleocapsid and (2) a cell-surface staining assay that measures antibody binding to GFP-tagged Spike expressed on the surface of CEM.NKr CCR5+ cells (S-CEM cells). We detected significantly higher IgG titers against Spike, RBD, and NTD in infants versus mothers by MSD ($p = 0.002$, 0.001, and 0.001, respectively, Fig. 1A–C), but there was no statistically significant difference in IgG titer against nucleocapsid ($p = 1.0$, Fig. 1D). When we restricted the analysis to infant-mother pairs ($N = 9$), significantly higher concentrations of binding antibodies were likewise observed in infants across antigens, except for nucleocapsid, as previously observed in the aggregated group (Fig. S2A–D). When we compared levels of cell surface staining, which measures levels of binding IgG antibodies to membrane-bound Spike, the infant response was significantly higher than the response in mothers ($p = 0.009$, Fig. 1E, S2E, and S3). Antibody binding to full-length Spike was correlated between methods indicating these assays are consistent metrics of the antibody binding response to Spike ($r = 0.7$; $p < 0.0001$, Fig. 1F). Antibody binding comparisons that were statistically significant remained significant after stratifying for HIV status (Table S1) and after stratifying for asymptomatic status (Table S2). Significance was lost when comparing infants and mothers with symptoms, likely due to the low number of symptomatic infants in the cohort ($N = 3$) (Table S2).

### Infants and mothers develop antibodies targeting the fusion peptide and stem helix of the S2 subunit

Our data suggested that levels of antibodies specific to Spike, RBD, and NTD were higher in infants versus mothers. To delineate binding sites with higher resolution and to identify epitopes outside of these domains, we used a previously described phage-based immunoprecipitation approach (Phage-DMS)[13,39] to map linear epitope binding profiles in plasma samples from mothers and their infants. Phage-DMS detects epitopes based on the enrichment of antibody-bound peptides expressed by T7 phage, and further defines mutations that confer escape by evaluating the loss of antibody binding to mutated peptides. The peptide library consisted of 39-amino acid peptides, tiled at single

## Table 1 | Participant age, HIV status, and disease severity

| | N total | Age: median (range) | N living with HIV or exposed (%) | N asymptomatic (%) | N mild symptoms (%) |
|---|---|---|---|---|---|
| Infants | 14 | 47.4 (8.7–80.9) weeks | 8 (57) | 11 (79) | 3 (21) |
| Mothers | 36 | 30 (20–38) years | 20 (56) | 29 (80) | 7 (20) |

Due to limited plasma availability, some samples were excluded from specific analyses. All N values for each individual assay are listed in associated figure legends. Mild symptoms were defined using the United States Centers for Disease Control definitions.

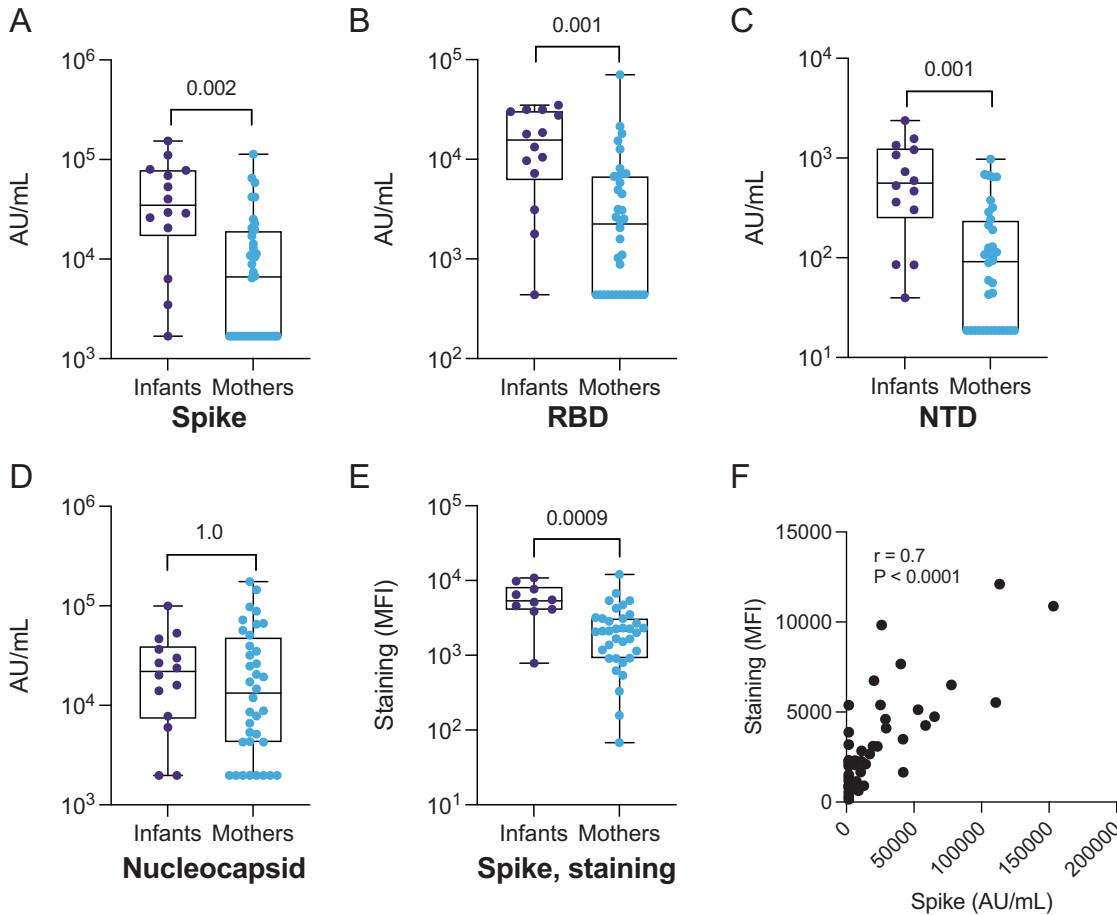

**Fig. 1 | IgG binding to SARS-CoV-2 antigens and S-CEM cell surface staining in SARS-CoV-2-seropositive infants and mothers.** IgG antibody binding titers to **A** full-length Spike, **B** RBD, **C** NTD, and **D** nucleocapsid in convalescent plasma from infants (purple, *N* = 14) and mothers (blue, *N* = 35) measured by commercial multiplexed electrochemiluminescent assay (MSD). **E** Antibody binding to Spike expressed on the CEM cell surface in infants (purple, *N* = 10) and mothers (blue, *N* = 35). The Spike staining was repeated in triplicate. **F** Spearman correlation coefficient and *p* value ($p = 2.5 \times 10^{-7}$) calculated for Spike IgG binding by MSD assay versus S-CEM cell surface staining. Box plots show the median center line and 25/75 percentiles. Whiskers show min and max values. *P* values are indicated above comparisons. Two-tailed Wilcoxon rank-sum test was used for all comparisons. MFI mean fluorescence intensity, AU/mL arbitrary units/milliliter.

amino acid intervals across B lineage Spike (Wuhan-Hu-1 sequence plus D614G), and included wildtype (i.e., Wuhan-Hu-1 plus D614G) sequences as well as every possible amino acid mutation at the central position of each peptide (see Methods).

We first mapped antibody binding to wild-type Spike sequences to determine linear epitope profiles in SARS-CoV-2-seropositive mothers and infants. Antibody binding to the FP and the SH-H, both in the S2 subunit, were the predominant responses in both infants and mothers (Fig. 2A). We confirmed that these regions were predominant and defined the residues involved in the binding response using principal component analysis (Fig. 2A and S4). Responses to FP and SH-H mirror previously identified epitopes in SARS-CoV-2-infected, unvaccinated individuals with mild COVID-19[13,19,20,40–42]. Linear responses to the NTD and the C-terminal domain (CTD), sometimes found in individuals hospitalized with COVID-19 or vaccinated individuals[19,20,40], were absent in both infants and mothers, consistent with the absence of vaccination or severe clinical manifestations of COVID-19 in this cohort. Responses to the RBD were absent as well, possibly because RBD epitopes can be conformational[43,44], while Phage-DMS captures only linear, non-glycosylated epitopes.

While the overall pattern of Spike antibody binding was focused on the FP and SH-H in both infants and mothers, we observed differences in the magnitude of enrichment between individuals. To test whether the magnitude of antibody enrichment in the FP and SH-H regions was different between infants and mothers in the aggregate, we summed the antibody enrichment at each position across the FP epitope (residues 805–835) and SH-H epitope (residues 1135–1170). Interestingly, infants had significantly higher summed enrichment in the FP than mothers ($p = 0.01$, Fig. 2B), while we observed no difference between infants and mothers in the SH-H region ($p = 0.8$, Fig. 2C). These results were consistent in the group of mothers and infants living with and exposed to HIV (Table S1) and when comparing asymptomatic mothers and infants (Table S2); however, significance was lost for the FP comparison when data were subset to the unexposed infants versus HIV-uninfected mothers, or symptomatic infants versus mothers, likely due to the smaller sample size upon stratification ($N = 4$ and $N = 3$, respectively).

## Mutations in Spike that lead to antibody binding escape in infants and mothers

Our phage-DMS library included sequences with all possible mutations to Spike at the central position of each phage-displayed peptide, allowing us to assess the impact of mutations on antibody binding—i.e., the ability of a mutated sequence to escape binding—in infants and mothers. To calculate the impact of a given mutation on antibody binding, we used a previously defined metric termed "scaled differential selection", defined as the log fold-change of antibody binding

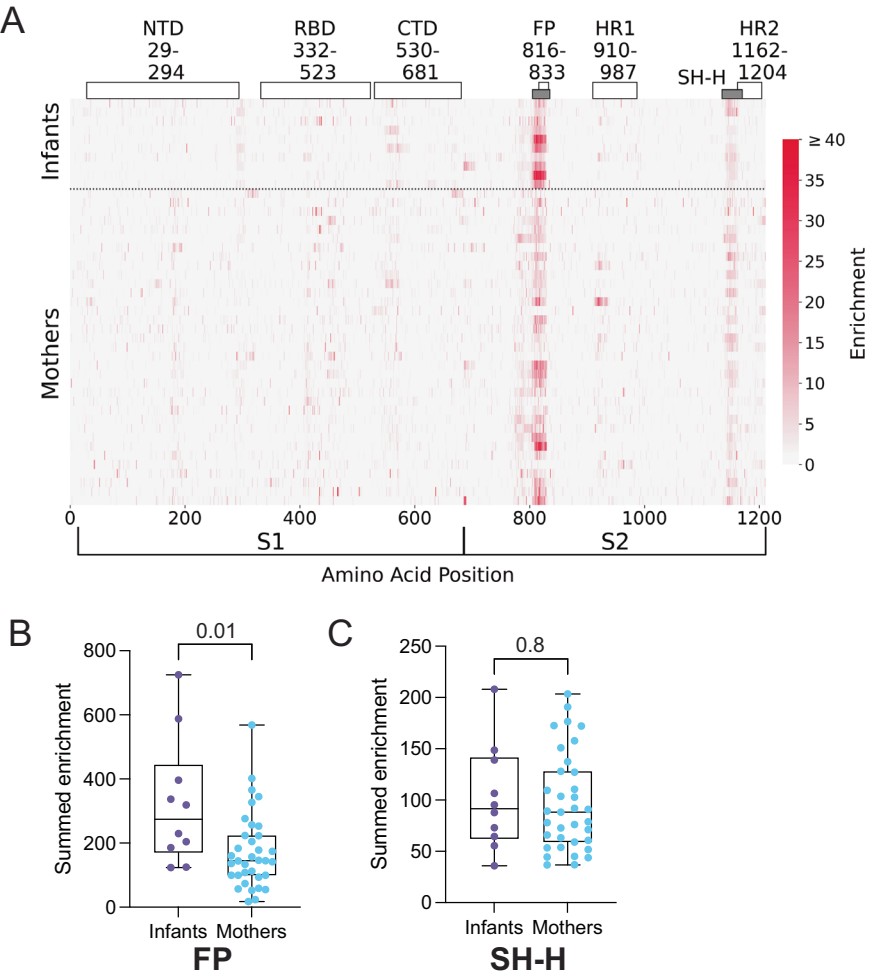

**Fig. 2 | Enrichment of plasma antibodies bound to wild-type Spike peptides.**
**A** Heatmap depicting peptide-bound, enriched antibody responses for individual infants ($N = 10$) and mothers ($N = 35$) (rows). Spike subdomains and amino acid ranges are labeled at the top. Gray bars indicate epitopes of focus in this study (FP and SH-H). Spike amino acid positions and S1 and S2 subunits are indicated at the bottom x-axis. The enrichment scale is indicated on the right. Summed antibody enrichment among mothers (blue, $N = 35$) and infants (purple, $N = 10$) in the FP epitope (**B**) and SH-H epitope (**C**). Box plots show the median center line and 25/75 percentiles. Whiskers show min and max values. *P* values are indicated above comparisons. Two-tailed Wilcoxon rank-sum test was used for both comparisons. NTD n-terminal domain, RBD receptor binding domain, CTD c-terminal domain, FP fusion peptide, HR1 heptad repeat 1, SH-H stem helix-heptad repeat 2, HR2 heptad repeat 2.

enrichment to the mutated sequence divided by the wild-type sequence at any given amino acid position (see Methods)[39]. Mutations that lead to a loss in antibody binding compared to the wild-type sequence result in a negative scaled differential selection value and mutations that lead to a gain in antibody binding result in a positive scaled differential selection value. Using this method, we calculated scaled differential selection for all possible mutations to Spike in infants and mothers. Because wild-type antibody enrichment in infants and mothers was isolated to the FP and SH-H of the S2 subunit in our previous analysis, we focused our attention on those regions for additional analysis.

In infants, a core set of mutations led to antibody binding escape, centered around residues 814–819, spanning the S2' transmembrane protease site 2 (TMPRSS2)-mediated cleavage site[45], with less pronounced escape downstream from that window in some infants (Fig. 3A). Infants appeared to have highly consistent escape profiles, suggesting infants may develop a convergent and/or less differentiated immune response to the FP. There was more variability in the escape profiles between individual mothers than between individual infants, but residues 813–820, encompassing the core positions seen in infants, were common sites of escape in mothers. However, we

observed more pronounced differential selection at residues 814 and 816–818 in infants than in mothers (Fig. 3B and S5). There were also some positions just upstream or downstream of this core sequence that were selected for escape in some mothers, particularly around positions 810 and positions 825–830. Additionally, we observed variability when comparing escape profiles of mothers directly to their infant in the FP region (Fig. 3A, B, dashed lines show infant-mother pairings).

To evaluate quantitatively whether escape profiles were more consistent among infants than among mothers, we used a previously described method to calculate escape similarity scores between two escape profiles[46]. This approach is akin to an optimal transport calculation, in which amino acid similarity dictates the "cost" associated with transitioning from one escape profile to the next (see Methods). Using this method, we calculated escape similarity scores at each amino acid position in the FP region and found that infants had higher scores at several residues within the epitope (Fig. 3C). Interestingly, the median similarity scores for each pairwise infant-infant comparison were higher than the infant-mother similarity scores suggesting there was more consistency within the infant group than within mother-infant pairs (Fig. 3D).

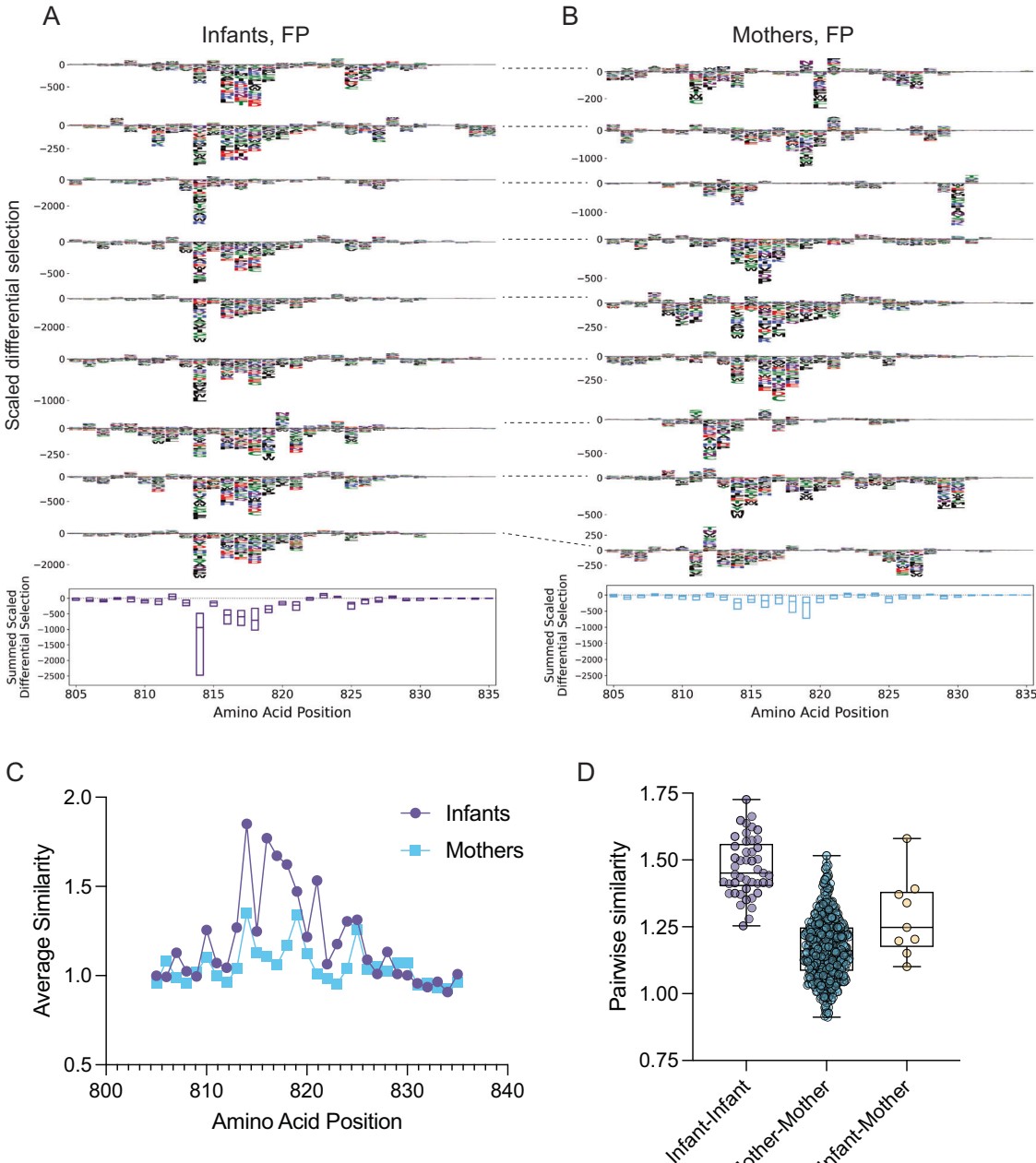

**Fig. 3 | Mutations that confer antibody binding escape in the Spike FP, and similarity in escape profiles. A** Individual infant ($N = 10$) escape profiles in the FP region (top), summed scaled differential selection at each position across all infant profiles (bottom). **B** Individual mother ($N = 35$) escape profiles in FP region (top), summed scaled differential selection at each amino acid position across all mother profiles (bottom). Dashed lines connecting escape profiles in (**A**) and **B** signify mother-infant pairs. **C** Average similarity score at each FP amino acid position for infants (purple) and mothers (blue). **D** Pairwise similarity scores across FP region among infants (purple, $N = 45$ infant-infant pairs), among mothers (blue, $N = 595$ mother-mother pairs), and for infant-mother pairs (yellow, $N = 9$ pairs). Summed scaled differential selection values for mothers (*B*, bottom panel) include data from all mothers (see additional logo plots in Fig. S5). Box plots show the median center line and 25/75 percentiles. Whiskers show min and max values.

Like the FP, several mutations led to a decrease in antibody binding in the SH-H epitope in both infants and mothers, but the escape profile spanned a wider range of amino acids than observed in the FP region (Fig. 4A, B and S6). In infants, mutations to amino acid 1152 led to the most negative median summed differential selection, suggesting it is commonly required for antibody binding to the SH-H (Fig. 4A). Conversely, mutations to residue 1149 most consistently led to antibody binding escape in mothers, suggesting plasma antibodies in infants and mothers vary in the degree of binding sensitivity to specific mutations (Fig. 4B and S6).

Additionally, we found the infants had more similar escape profiles at several positions in the SH-H versus mothers (Fig. 4C), and that median infant-infant, infant-mother, and mother-mother pairwise similarity scores differed, although to a lesser extent compared to the FP region (Fig. 4D).

**Neutralization and ADCC activity among infants and mothers**
We hypothesized that higher antibody binding levels against full-length Spike, RBD, NTD, and FP in infants might correlate with elevated functional antibody activity, including neutralization

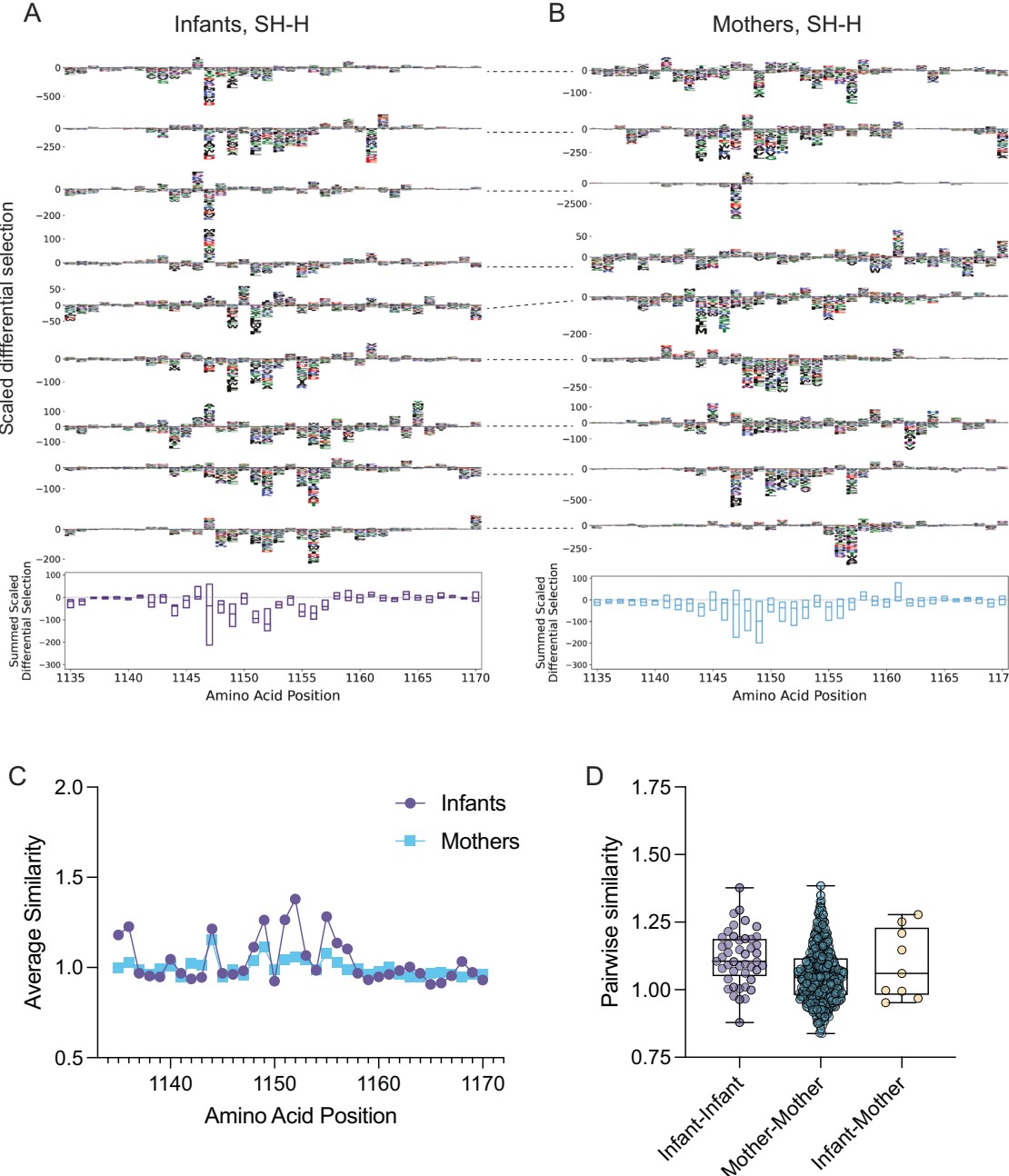

**Fig. 4 | Mutations that confer antibody binding escape in the Spike SH-H, and similarity in escape profiles. A** Individual infant ($N = 10$) escape profiles in the SH-H region (top), summed scaled differential selection at each amino acid position across all infant profiles (bottom). **B** Individual mother ($N = 35$) escape profiles in FP region (top), summed scaled differential selection at each position across all mother profiles (bottom). Dashed lines connecting escape profiles in (**A**) and (**B**) signify mother-infant pairs. **C** Average similarity score at each SH-H amino acid position for infants (purple) and mothers (blue). **D** Pairwise similarity scores across SH-H among infants (purple, $N = 45$ infant-infant pairs), among mothers (blue, $N = 595$ mother-mother pairs), and for infant-mother pairs (yellow, $N = 9$ pairs). Summed scaled differential selection values for mothers (**B**, bottom panel) include data from all mothers (see additional logo plots in Fig. S6). Box plots show the median center line and 25/75 percentiles. Whiskers show min and max values.

and/or ADCC. We, therefore, tested plasma in a Spike-pseudotyped lentiviral neutralization assay[47,48]. Mothers living with HIV and HIV-exposed infants were excluded from the neutralization analysis because of the presence of ART in plasma samples, which would inhibit infection in the assay. Interestingly, we found no significant difference in neutralization titer between mothers and infants in analyses that included all mothers without HIV and unexposed infants ($p = 0.9$, Fig. 5A), nor in paired mothers and infants ($p = 0.7$, Fig. 5B), suggesting that, in the

context of this limited sample size, higher Spike antibody titers in infants do not directly translate to higher levels of neutralization. Additionally, among pooled infants and mothers, neutralization titers did not correlate with Spike binding measured by MSD ($r = 0.3$, $p = 0.2$, Fig. S7A), but there was an association with binding measured by S-CEM surface staining ($r = 0.5$, $p = 0.04$, Fig. S7B).

To test whether there were differences in ADCC activity between mothers and infants, we used an established flow cytometry-based

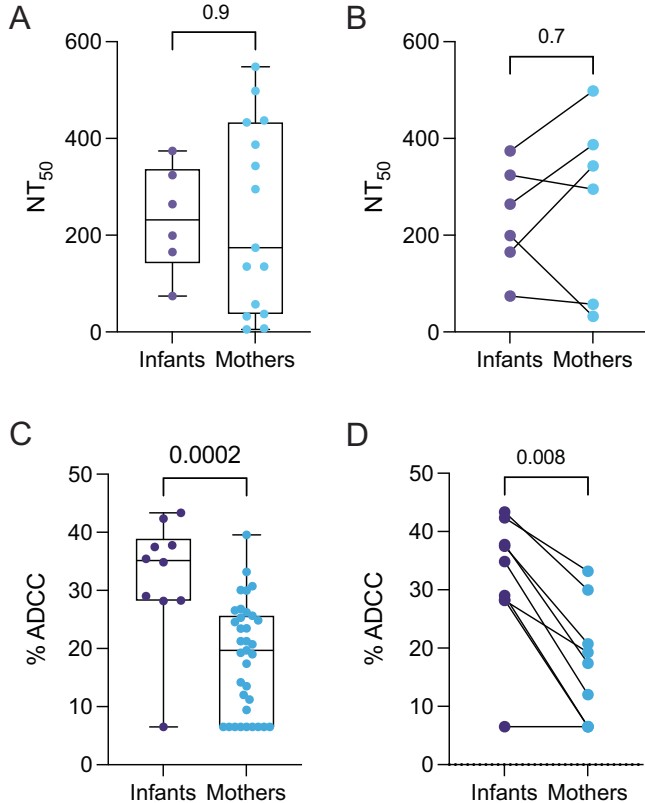

**Fig. 5 | Pseudovirus neutralization and ADCC activity in SARS-CoV-2-seropositive infants and mothers.** 50% neutralization titers (NT$_{50}$) against Wuhan-Hu-1 S-pseudotyped lentiviral particles in **A** all infants (purple, $N = 6$) and mothers (blue, N = 15) and **B** matched mother-infant pairs ($N = 6$). Neutralization activity was measured in duplicate or triplicate. Plasma ADCC activity in **C** all infants ($N = 10$) and mothers ($N = 35$), and **D** matched mother-infant pairs ($N = 9$). Plasma ADCC activity was measured in triplicate. Box plots show the median center line and 25/75 percentiles. Whiskers show min and max values. *P* values are indicated above each comparison. Unmatched comparisons: two-tailed Wilcoxon rank-sum test; matched comparisons: two-tailed Wilcoxon matched-pairs sign-rank test.

## Discussion

Antibody responses to viral infection and vaccination can differ between infants and adults. However, studies of binding and functional antibody responses to SARS-CoV-2 in infants are rare, as are direct comparisons between infants and adults in the context of SARS-CoV-2 infection. Understanding these differences could inform efforts to treat and prevent COVID-19 across the age spectrum. In this study, we observed significant differences in antibody binding and functional activity in plasma from SARS-CoV-2-seropositive women and their infants within a single cohort. Notably, infants had higher levels of ADCC and Spike-binding antibodies, but not neutralizing antibodies. We also observed differences in the patterns of escape for infant antibodies targeting the FP epitope compared to mothers.

Levels of ADCC activity were significantly higher in infants than in mothers, suggesting infants develop either more abundant ADCC antibodies or antibodies with more ADCC potency during SARS-CoV-2 infection. ADCC activity has been linked to protection against SARS-CoV-2[27,29,51] and other viruses, including HIV[31–34]. Whether the higher levels of ADCC observed in infants contributes to protection or other aspects of pathogenesis is an important area for future study.

In addition to having higher levels of Spike-specific ADCC activity, infants displayed higher levels of antibody binding to full-length SARS-CoV-2 Spike protein than mothers. Infants also had higher levels of antibodies to RBD, NTD, and FP, but not nucleocapsid or SH-H, suggesting a more abundant or high-affinity humoral immune response to specific Spike subdomains in infants. The observation that IgG titers against nucleocapsid and SH-H were the same in infants and adults in this study suggests that the stronger responses in infants to other domains cannot simply be explained by higher overall B lymphocyte levels reported in infants versus adults[52]. Previous studies have detected lower[53], higher[22,24], or equivalent[23,25] levels of antibody binding to Spike or its subdomains in pediatric cohorts compared to adults. One possible reason for this variation is the inclusion of children who span a wide age range in prior studies, whereas this study specifically focused on infants (under 19 months of age), with samples collected within a single cohort study[35]. Notably, despite reported differences in the development of B-cell responses in people living with HIV[54], antibody binding and functional responses observed in infants generally remained significantly elevated after stratifying for HIV status.

Our epitope mapping experiments demonstrated that antibody binding to wild-type peptides was common at the FP and SH-H regions in both infants and mothers, with elevated responses to FP in infants. Further, antibody escape profiles were more similar among infants than adults, particularly in the FP region. FP escape profiles were more similar among infant-infant pairings than among infants paired with their mothers, suggesting age may be a more important indicator of antibody escape pathways than the specific genetics of the immune response in an individual. Overall, these results are suggestive of a convergent immune response to the FP in infants and may indicate that infant FP antibody lineages develop differently than in adults. Given the more robust response to FP in infants and the elevated levels of ADCC in infants, it may be informative to isolate FP-specific monoclonal antibodies from infants and assess whether they have Fc-mediated effector functions in future studies.

Interestingly, though we observed elevated Spike protein and RBD antibody binding in infants, and the RBD is the main target of neutralizing antibodies against SARS-CoV-2[15], neutralization titers were similar between infants and mothers. Studies comparing neutralization activity in infants and adults are rare, and like studies of SARS-CoV-2 Spike antibody binding, there is variability among analyses of pediatric versus adult SARS-CoV-2 neutralization across different cohorts. One very small study of infants (<3 months of age, $N = 4$) and their parents reported a modest increase in neutralizing antibody titers

cellular assay that measures the level of ADCC when S-CEM cells are exposed to both plasma and peripheral blood mononuclear cells (PBMCs) from healthy individuals as effector cells (Fig. S8)[49,50]. Infants had significantly higher ADCC activity than mothers when compared in aggregate ($P = 0.002$, Fig. 5C), and activity remained significantly elevated in infants upon stratifying for HIV status (Table S1), or when comparing asymptomatic mothers and infants (Table S2). When we compared infant-mother pairs only, which reduced the total number of participants to similar numbers of individuals tested in the neutralization assay, the levels of ADCC activity in infants remained significantly higher than in mothers ($P = 0.008$, Fig. 5D), suggesting infants and mothers have different levels of SARS-CoV-2 antibody functional activity.

Among all participants, ADCC activity was associated with Spike, RBD, and NTD antibody binding by MSD ($r = 0.7$, $p = <0.0001$; $r = 0.6$, $p = <0.0001$; $r = 0.6$, $p = <0.0001$, respectively, Fig. S7C−E) and S-CEM cell surface staining ($r = 0.9$, $p < 0.0001$, Fig. S7F). ADCC activity was not correlated with neutralization titer, highlighting the differences in antibody function in these plasma samples ($r = 0.2$, $p = 0.4$, Fig. S7G). Additionally, ADCC activity was moderately associated with FP summed enrichment ($r = 0.5$, $p = 0.0005$, Fig. S7H), but not SH-H summed enrichment ($r = 0.2$, $p = 0.2$, Fig. S7I) in aggregated infants and their mothers, suggesting a possible role for FP antibodies in mediating ADCC activity.

in infants[25], and another study (0–4 years of age, N = 15) found two-fold higher levels of neutralization in infants[22]. In a study of older children aged 3–11 years versus adults, neutralization was comparable against ancestral Wuhan-Hu-1 SARS-CoV-2 and multiple variants of concern[21].

Several factors may contribute to conflicts in reported antibody binding and neutralization activity between pediatric cohorts and adults. Differences in binding and neutralization assay methodology could contribute to study-to-study variability, as well as variation in median cohort age, immune history, and sample timing in relation to infection, which varies across studies. While infants and mothers in this study were sampled within a defined window at approximately three-month intervals, decay in antibody titers can occur during this period[55]. However, the variation due to decay is likely similar within infants and mothers because there was no significant difference in estimated time since infection between the two groups. Additionally, rates of antibody decay were found to be similar among mothers and infants in this cohort in a previous study[35].

Overall, these results suggest that SARS-CoV-2 plasma antibody binding, escape pathways, and functional capacity differ between infants and adults infected with SARS-CoV-2. Mechanistic drivers of these differences are likely complex and could include differences in immune developmental stage, varying pathogen exposure history, and possible differences in viral load among infants and mothers that should be explored in future work. This study also raises the question of whether there are analogous differences in the response to vaccination in infants. Overall, the finding that infants develop higher levels of binding and ADCC antibodies compared to adults should motivate evaluation of these activities in pathogenesis, particularly in cohorts with a broader spectrum of disease severity, given documented age-related differences in COVID-19 severity[6–8], and provides an important baseline for evaluating and designing vaccine and antibody-based therapeutic options across the age spectrum.

## Methods
### Study participants
Mother-infant pairs enrolled in an existing prospective cohort of mother-to-child virome transmission in Nairobi, Kenya (the Linda Kizazi study) provided written informed consent for participation in the parent cohort study, which included blood sample collection and immunology studies; an additional written informed consent was collected for SARS-CoV-2 testing. Mothers and infants attended clinic visits approximately every 3 months, at which time clinical data and samples, including blood were collected. The first SARS-CoV-2-seropositive plasma samples from mothers (N = 36) and infants (N = 14) that seroconverted to SARS-CoV-2 based on Nucleocapsid ELISA between April 2019-December 2020, as reported in ref. 35, were included in this study. Estimated time since infection was calculated as the midpoint between the last seronegative sample and the first seropositive sample, unless the last negative sample was before May 1, 2020 (which we use as an estimate of the beginning of the risk period for SARS-CoV-2 infection in Kenya), in which case May 1, 2020, was used as the date of the last seronegative sample. Infant sex was recorded, but not considered in the study design due to the limited number of participants. The Kenyatta National Hospital-University of Nairobi Ethics and Research Committee (P472/07/2018), the CHUM Research Center, and the University of Washington and Fred Hutchinson Cancer Center Institutional Review Boards (STUDY00004006) approved all human participant study procedures.

### Multiplexed chemiluminescent antibody binding assay
SARS-CoV-2 Spike, RBD, NTD, and nucleocapsid IgG antibody levels were detected using a commercially available multiplexed chemiluminescent binding assay (V-PLEX COVID-19 Coronavirus Panel 2 (IgG) Kit, Cat. No. K15369U-2, Mesoscale Diagnostics, MSD). Plasma samples were heat-inactivated for 60 min at 56 °C and diluted 1:5000

according to the manufacturer's instructions. Diluted samples and manufacturer-provided calibrator and control samples were applied to blocked 96-well assay plates and incubated for 2 h at room temperature (RT). Plates were washed and incubated with detection antibody for 1 h After the addition of MSD GOLD Read Buffer B, plates were immediately read on the MESO QuickPlex SQ 120MM instrument connected to Methodical Mind software (Mesoscale Diagnostics) using default parameters. Raw data were processed in Discovery Workbench software version 4.0 (Mesoscale Diagnostics). Sample intensity was converted to Arbitrary Units/mL (AU/mL) based on the calibrator standard curve included in the assay kit as part of the Discovery Workbench workflow. Antibody concentrations for all SARS-CoV-2 antigens were above the calculated lower limits of detection. As previously mentioned, only samples that were positive by SARS-CoV-2 Nucleocapsid ELISA were included in the study. In addition, a within-assay positivity threshold was set for each SARS-CoV-2 antigen in the MSD assay by measuring the mean AU/mL plus three standard deviations above the mean among a population of 18 pre-pandemic samples collected as part of the Linda Kizazi study[56]. Measurements in the experimental (SARS-CoV-2 positive) population that fell below this threshold were set to the midpoint between zero and the threshold.

### Cell surface antibody staining
CEM.NKr CCR5 (parental) cells were originally sourced from HIV Reagent Program (cat #4376) and were authenticated by STR profiling. CEM.NKr CCR5-Spike (Wuhan-Hu-1 isolate, S-CEM) cells were derived from CEM.NKr CCR5 cells and Spike expression was confirmed by cell-surface staining and flow cytometry[49]. About 300,000 parental or S-CEM cells were stained with plasma (1:500 final dilution) or control mAbs (1 μg/mL final concentration) for 45 min at RT. Cells were washed twice with PBS and 100 μL of Goat anti-human IgG (H + L) Alexa647 secondary antibody (2 μg/mL, Invitrogen #A-21445) and Aqua viability dye (Thermo Fisher Scientific, Cat# L34957) was added for 20 min at RT. After staining, cells were washed twice with PBS and fixed with 2% formaldehyde in PBS. Cells were acquired on an LSRII instrument (BD Biosciences), with 10,000 live cell events recorded per sample. The gating strategy for cell surface staining is shown in Fig. S3. Cells were identified according to cell morphology by light-scatter parameters and excluding doublets cells. Dead cells (Aqua[High]) were then excluded. Finally, the GFP+ cells were used to detect and measure the Spike-specific antibodies present in the plasma. Data analysis was performed using FlowJo v10.7.1 (TreeStar). Spike-specific surface antibody staining was defined as: (Alexa647 MFI of Live GFP[+] S-expressing cells + plasma/antibody) – (Ax647 MFI of Live GFP[-] parental cells + plasma/antibody). The mean of the mean fluorescence intensity (MFI) of SARS-CoV-2 seronegative plasma samples plus three standard deviations was used as a positivity threshold and staining measurements below that threshold were set to the midpoint between zero and the threshold.

### SARS-CoV-2 spike phage-deep mutational scanning (phage-DMS)
The composition, preparation, and use of the T7 phage display library used in this study has been described previously in refs. 13,40. To probe plasma samples, the phage library was diluted with Phage Extraction Buffer (20 mM Tris-HCl, pH 8.0, 100 mM NaCl, 6 mM MgSO₄) to 4.96 × 10⁹ plaque-forming units/mL, to account for ~200,000-fold representation of all 24,820 peptides included in the library. One mL of diluted library was incubated with 10 μL of heat-inactivated (56 °C for 60 min) plasma and incubated overnight in 1.1 mL deep 96-well plates (Costar) at 4 °C with rocking. A 1:1 mixture of Protein A and Protein G Dynabeads (Invitrogen) was prepared and 40 μL of the mixture was added to each well. The plate was incubated again at 4 °C for 4 h with rocking. Dynabeads bound to antibody-phage

complexes were isolated using a magnet, washed three times with 400 µL Wash Buffer (150 mM NaCl, 50 mM Tris-HCl, 0.1% (v/v) NP-40, pH 7.5), and resuspended in 40 µL of water. Bound phage particles were lysed by incubating resuspended samples at 95 °C for 10 min. To evaluate the starting frequencies of peptides in the library, the original diluted phage library (not incubated with plasma) was also lysed in parallel. Plasma samples were tested in technical duplicates on separate days.

Phage DNA was subjected to two rounds of PCR using Q5 High-Fidelity 2X Mastermix (NEB). In the first round, 10 µL of the lysed phage was used as a template with R1_FWD (TCGTCGGCAGCGTCTCCAGT CAGGTGTGATGCTC) and R1_REV (GTGGGCTCGGAGATGTGTATAA GAGACAGCAAGACCCGTTTAGAGGCCC) primers in a 25 µL reaction volume. For the second round PCR, 2 µL of the Round 1 reaction was added to unique dual-indexed barcoding primers R2_FWD (AATGA TACGGCGACCACCGAGATCTACACxxxxxxxxTCGTCGGCAGCGTCTC CAGTC) and R2_REV (CAAGCAGAAGACGGCATACGAGATxxxxxxxxG TCTCGTGGGCTCGGAGATGTGTATAAGAGACAG), where "xxxxxxxx" corresponded to a unique 8-nt indexing sequence. Products were quantified using the Quant-iT Pico Green Kit (Thermo Fisher) according to the manufacturer's instructions. Samples were pooled in equimolar quantities, and the input library sample was included at tenfold molar excess. The final pool was gel-purified, quantified using the KAPA Library Quantification Kit (Roche), and submitted for sequencing on an Illumina HiSeq using 125 base pair single-end reads.

## Phage-DMS data analysis

The *phippery* software framework (https://matsengrp.github.io/phippery/) was used to analyze phage-DMS sequencing data. First, sample reads were processed into peptide counts in a Nextflow[57] pipeline that uses *Bowtie2*, v2.4.2[58], for short-read alignment and *Samtools*, v1.3[59], to gather sequencing statistics. Peptide counts from all samples were collected into a *xarray*, v0.16.1[60] dataset, merging sample and peptide metadata with their respective count. Peptide enrichment and differential selection were computed using Python (v3.6.12) modules provided in *phippery*.

## Escape profile similarity scoring

The comparison of escape profiles was conducted as described previously in ref. 46. Additional details can be found at https://matsengrp.github.io/phippery/esc-prof.html.

## SARS-CoV-2 Spike-pseudotyped lentivirus production

Pseudovirus expressing SARS-CoV-2 Spike protein was produced and titered using established methods[47]. HEK293T cells were obtained from ATCC (cat #CRL-3216) and were authenticated by STR profiling. HEK293T cells were seeded at a density of $5 \times 10^5$ cells per well in complete DMEM (10% fetal bovine serum, 2 mM L-glutamine, and penicillin/streptomycin/fungizone) in six-well dishes. After 16–23 h, cells were transfected using FuGene-6 (Promega #E2692) with the following constructs: the Luciferase_IRES_ZsGreen backbone, Gag/Pol, Rev, and Tat lentiviral helper plasmids, and plasmid HDM_Spikedelta21 containing the codon-optimized Spike sequence from the Wuhan-Hu-1 strain and a 21 amino acid deletion in the cytoplasmic tail[36]. After 25 h, the media was replaced with fresh complete DMEM. After 50–60 h post-transfection, viral supernatants were collected, filtered through 0.22-µm Steriflip filters, concentrated and stored at −80 °C. To titer pseudovirus, $1.25 \times 10^4$ HEK293T-ACE2 cells were seeded in 96-well black-walled plates, and 100 µL of serially diluted viral supernatant was added per well in duplicate 16–24 h later. VSV-G and no viral entry protein (VEP) positive and negative control wells were included. After 60 h, 100 µL supernatant was removed from each well and 30 µL of Bright-Glo (Promega #E2620) was added. Relative luciferase units were measured using a LUMIstar Omega plate reader (BMG Labtech) equipped with Omega software (v5.50

R4). HEK293T-ACE2 cells were kindly provided by Dr. Jesse Bloom, and ACE2 expression was previously confirmed using anti-human ACE2 polyclonal goat IgG[47].

## 384-well format SARS-CoV-2 Spike neutralization assays

SARS-CoV-2 Spike-pseudotyped lentiviral neutralization assays were conducted in a 384-well plate format as described previously in ref. 48. Briefly, black-walled, clear bottom, poly-L-lysine-coated 384-well plates (Thermo Scientific #142761) were seeded with $3.75 \times 10^3$ HEK293T-ACE2 cells (BEI Resources, NR-52511) per well in 30 µL of complete DMEM. After 12–16 h, plasma samples were serially diluted threefold in complete DMEM starting at 1:20, for a total of six dilutions. Spike-Δ21 pseudotyped lentiviral particles were diluted 1:5 and added to diluted plasma samples at an equal volume. Pseudovirus and plasma were incubated for 1 h at 37 °C, and 30 µL of the virus-plasma samples were added to cells. Plasma-free wells containing only virus and cells were included as negative controls.

After 55 h, luciferase activity was measured using the Bright-Glo Luciferase Assay System (Promega E2610) on a LUMIstar Omega plate reader (BMG Labtech) equipped with Omega software (v5.50 R4). Fraction infectivity was calculated by dividing the mean RLU from each plasma dilution sample by the mean of the plasma-free (virus plus cells only) wells. The plasma dilution that inhibited infection by 50% ($IC_{50}$) was calculated in Prism by fitting fraction infectivity to a Hill curve with a bottom and top constrained to 0 and 1, respectively, and $IC_{50}$ constrained to >0. The $NT_{50}$ for each plasma sample was calculated as the reciprocal of the $IC_{50}$. Plasma samples with undetectable neutralization activity were assigned an $NT_{50}$ of 20, which was the lower limit of the plasma dilution series.

All samples were assayed in technical duplicate and additional replicate experiments were conducted using separately transfected pseudovirus and freshly thawed cells. If there was a greater than threefold difference between two replicate $NT_{50}$ values, we included that plasma sample in a third replicate, except for a single mother, for whom the sample was not available. As such, $NT_{50}$ values reported are the mean of at least two or three replicates.

## SARS-CoV-2 Spike ADCC assay

SARS-CoV-2 Spike glycoprotein-specific ADCC activity was measured against CEM.NKr CCR5+ cells stably expressing GFP-tagged S protein (Wuhan-Hu-1 isolate, S-CEM cells)[49,50]. S-expressing cells were mixed at a 1:1 ratio with parental CEM.NKr CCR5+ cells (HIV Reagent Program #4376) and the target cell mixture was labeled with Aqua viability dye (Thermo Fisher Scientific, Cat# L34957) and eBioScience eFluor670 cell proliferation dye (Thermo Fisher Scientific, Cat#65-0840-85). In parallel, PBMCs from healthy uninfected adult individuals were labeled with eBioScience eFluor450 cell proliferation dye (Thermo Fisher Scientific, Cat#65-0842-85) after overnight rest to use as effectors in the assay. PBMCs from a single donor were used in all replicate experiments. Labeled target and effector cells were added to 96-well V bottom plates at a 1:10 ratio. Plasma (1:500 final dilution) or control monoclonal antibodies (1 µg/mL final concentration) were added to corresponding wells and wells were mixed by pipetting up and down. Plates were then centrifuged for 1 min at 300×*g* to bring the cells into close association. ADCC was allowed to occur for 5 h at 37 °C, after which cells were fixed in 2% formaldehyde in PBS. Cells were acquired on an LSRII instrument (BD Biosciences) using built-in BD FACSDiva software (v6), with 10,000 Live eFluor670⁺ eFluor450⁻ target cell events recorded per sample. The gating strategy for ADCC activity measurements is shown in Fig. S8. Target cells were identified according to cell morphology by light-scatter parameters and excluding doublets. Cells were then gated on eFluor670+ cells (excluding the effector cells labeled with eFluor450). Finally, the percentage of GFP+ target cells was used to calculate the ADCC activity. Data analysis was performed using

FlowJo v10.7.1 (TreeStar). ADCC activity was calculated using the following formula after gating on target cells: $100 \times [(\% \text{ GFP}^+ \text{ cells in targets plus effectors}) - (\% \text{ GFP}^+ \text{ cells in targets plus effectors plus plasma/antibody})]/(\% \text{ GFP}^+ \text{ cells in targets alone})$. The following mAbs were included as positive controls in each experiment: CR3022 (Abcam, cat# ab278112), CV3-1, CV3-13, CV3-25, and CV3-25 GASDALIE[50]. HIV-specific human monoclonal antibody 17b (produced in-house from publicly available sequences) and plasma from five SARS-CoV-2 seronegative individuals were included as negative controls. The mean %ADCC of SARS-CoV-2 seronegative plasma samples plus three standard deviations was used as a positivity threshold and ADCC measurements below that threshold were set to the midpoint between zero and the threshold.

### Additional statistical analyses

Wilcoxon rank-sum tests (also known as Mann–Whitney *U*-tests) or Wilcoxon matched-pairs signed rank tests were performed in Prism (v9, Graphpad). *P* values for binding data collected using the MSD immunoassay, for which several antigens were probed at one time, were corrected for multiple hypothesis testing post-hoc using the Bonferroni method, accounting for four hypotheses. *P* values <0.05 were considered significant.

### Reporting summary

Further information on research design is available in the Nature Portfolio Reporting Summary linked to this article.

## Data availability

The unprocessed phage-DMS sequencing data generated in this study have been deposited to the SRA under accession code PRJNA872509. The antibody binding, cell-surface staining, neutralization, and ADCC data generated in this study are provided in the Source Data file as part of the Supplementary Information. Source data are provided with this paper.

## Code availability

Code for data processing and visualization after read alignments is provided at https://github.com/ksung25/LK-SARS-CoV-2 and https://doi.org/10.5281/zenodo.8095491. The escape profile similarity scoring pipeline and associated documentation can be found at https://matsengrp.github.io/phippery/esc-prof.html.

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

## Acknowledgements

We thank the Linda Kizazi study team for their contributions and especially thank participating mothers and their infants for contributing samples. We thank the Genomics and Flow Cytometry core facilities and staff at Fred Hutchinson Cancer Center for assistance with data collection. We thank members of the Overbaugh and Matsen labs for helpful discussion and advice. This work was supported by NIH grants R01 AI138709 (PI Overbaugh) and R01 AI146028 (PI Matsen). Frederick Matsen is an investigator of the Howard Hughes Medical Institute and Scientific Computing Infra-structure at Fred Hutchinson Cancer Center, funded by ORIP grant S10OD028685. C.I.S. is the recipient of the NIH K99 award AI171000. Z.A.Y. is the recipient of the NIH F30 award AI165112. Work in the Finzi lab was supported by a CIHR foundation grant #352417, a CIHR operating Pandemic and Health Emergencies Research grant #177958, and an Exceptional Fund COVID-19 from the Canada Foundation for Innovation (CFI) #41027 to A.F. A.F. is the recipient of Canada Research Chair on Retroviral Entry no. RCHS0235 950-232424. G.B.-B. is the recipient of an FRQS doctoral fellowship. The Linda Kizazi cohort study was supported by grants from the Canadian Institutes of Health Research (COVID-19 May 2020 Rapid Research Funding Opportunity Operating Grant 202005, Project Grant 201709 to S.G.) and the US National Insti-tutes of Health (grant numbers R01HD092311 to D.A.L. and R00DK107923 to E.S.L.). The following reagent was obtained through the NIH HIV Reagent Program, Division of AIDS, NIAID, NIH: CEM.NKR CCR5+ Cells, ARP-4376, contributed by Dr. Alexandra Trkola.

## Author contributions

J.O., D.A.L., and C.I.S. conceived of the study and designed experiments, with support from A.F. and G.B.-B. for ADCC experiments. Software and sequencing data management was conducted by K.S. and J.G. under the supervision of F.A.M. Experiments and formal analysis were led by C.I.S. with additional support from K.S., H.W., Z.A.Y., and G.B.-B, under the supervision of J.O., D.A.L., A.F., and F.A.M. Cohort resources and man-agement was conducted by E.O., J.A., E.R.B., J.S., S.G., J.K., D.W., and D.A.L. The original manuscript draft was prepared by C.I.S., J.O., and D.A.L., and all others contributed to the editing and approval of the final manuscript.

## Competing interests

The authors declare no competing interests.

## Additional information

[1]Human Biology Division, Fred Hutchinson Cancer Center, Seattle, WA, USA. [2]Public Health Sciences Division, Fred Hutchinson Cancer Center, Seattle, WA, USA. [3]Medical Scientist Training Program, University of Washington, Seattle, WA, USA. [4]Centre de Recherche du CHUM, Université de Montréal, Montreal, QC, Canada. [5]Département de Microbiologie, Infectiologie et Immunologie, Université de Montréal, Montreal, QC, Canada. [6]Centre de Recherche du CHU Sainte-Justine, Université de Montréal, Montreal, QC, Canada. [7]Department of Pediatrics and Child Health, University of Nairobi, Nairobi, Kenya. [8]Department of Global Health, University of Washington, Seattle, WA, USA. [9]Department of Research and Programs, Kenyatta National Hospital, Nairobi, Kenya. [10]Howard Hughes Medical Institute, Chevy Chase, MD, USA. [11]These authors jointly supervised this work: Dara A. Lehman, Julie Overbaugh. ✉e-mail: dlehman@fredhutch.org; joverbau@fredhutch.org

