## [Peer Review File · Nature Communications]

Elevated binding and functional antibody responses to SARS-CoV-2 in infants versus mothersREVIEWER COMMENTS

Reviewer #1 (Remarks to the Author):

In this study, Stoddard et al. compared the antibody response between postpartum women and their infants infected with SARS-CoV-2. MSD and cell-surface staining assays showed that IgG titers against Spike, RBD, and NTD in infants were significantly higher in infants than in mothers. Phage-DMS results further demonstrated that infants have higher antibody levels against the FP region of the S2 subunit, and more consistent antibody escape profiles. The authors also showed that plasma ADCC activity is higher in infant, though the viral neutralization ability is similar between infants and mothers. Overall, study is well written and can potentially provide important insights into age-dependent severity of COVID-19. However, there are some concerns that need to be addressed.

Specific comments:

1. Can the author include the information about which viral variant infected the cohort?
2. Do the author know the number of days between viral infection and sample collection? The duration between viral infection and sample collection may influence the antibody titer of the sample (PMID: 33535236, PMID: 33778792), thus can be a confounding variable in this study.
3. Since disease severity is known to influence the magnitude of antibody response (PMID: 34330709), the authors may want to control for disease severity (i.e. mild symptom vs symptomatic) in their analyses.
4. Based on the ADCC results, the authors claimed that "infants develop unique functional antibody repertoires during SARS-CoV-2 infection" (lines 239-240). At the same time, ADCC activity has a very high correlation with S-CEM cell surface staining (Figure S5F) but not neutralizing activity (Figure S5G). So it seems like the higher ADCC activity in infants is simply due to a higher level of non-neutralizing antibody. To claim that infants and mothers have difference "functional repertoires", antibody level needs to be controlled for.

Reviewer #2 (Remarks to the Author):

This is a well-written concise manuscript that compares antibody responses in mothers and their infants who acquired COVID-19 after delivery thus precluding any complication of placentally derived infant antibodies. They used the first positive sample in both infants and mothers. Results demonstrate differences in antibody responses including higher levels of Abs targeting the fusion domain of S2 and ADCC responses but more negative scaled differential selection scores indicating loss of Ab function against viral escape mutants in infants versus adults. The manuscript would be greatly strengthened if the authors included data on clinical disease in the cohort.

Specific questions:

1. Does disease severity correlate with Ab responses either in infants or mothers?
2. Is there any information on viral shedding and does this contribute to differences in scaled differential selection. One might hypothesize that ongoing viral replication might select for viral escape mutants, which could impact Ab selection.
3. What is the mechanism by which age impacts antibody selection?
4. Since this was part of a longitudinal cohort study, do the authors have subsequent samples to see if the immune response evolves over time?

Reviewer #1 (Remarks to the Author):

In this study, Stoddard et al. compared the antibody response between postpartum women and their infants infected with SARS-CoV-2. MSD and cell-surface staining assays showed that IgG titers against Spike, RBD, and NTD in infants were significantly higher in infants than in mothers. Phage-DMS results further demonstrated that infants have higher antibody levels against the FP region of the S2 subunit, and more consistent antibody escape profiles. The authors also showed that plasma ADCC activity is higher in infant, though the viral neutralization ability is similar between infants and mothers. Overall, study is well written and can potentially provide important insights into age-dependent severity of COVID-19. However, there are some concerns that need to be addressed.

We are grateful to the reviewer for the positive feedback and address specific concerns below.

Specific comments:

1. Can the author include the information about which viral variant infected the cohort?

Stool samples collected during the study period were available for a small subset of participants and RNA-positive samples were previously subjected to whole-genome sequencing. Those samples were classified as SARS-CoV-2 B.1 lineage ("ancestral", includes D614G), and we have now included this information and the associated reference in the revised text (line 100).

2. Do the author know the number of days between viral infection and sample collection? The duration between viral infection and sample collection may influence the antibody titer of the sample (PMID: 33535236, PMID: 33778792), thus can be a confounding variable in this study.

This is an important concern, and we note a couple of features specific to our cohort related to this point: (1) testing resources were limited in Kenya during the study period compared to a place like the US where anyone with symptoms was generally able to be tested. For this reason, we don't know the exact timing of infection and can generate a best estimate based on the midpoint between the last seronegative and first seropositive plasma samples. (2) Because infants and mothers included in the study were part of the same cohort, with similar/identical sampling intervals, the influence of this limitation was the same across the two groups.

To further address this point, we have included an analysis of estimated time-since-infection using the midpoint described above. When we compare mothers versus infants there is no significant difference, suggesting the two groups are well-matched within the cohort. (Figure S1 and see lines 85-92). Thus, number of days between viral infection and sample collection does not explain the increased levels of antibody binding and ADCC responses we observed in the infants.

3. Since disease severity is known to influence the magnitude of antibody response (PMID: 34330709), the authors may want to control for disease severity (i.e. mild symptom vs symptomatic) in their analyses.

This is a very important point. Our cohort is limited in that most individuals were asymptomatic, or had mild disease—i.e. no severe cases which would be necessary to fully assess the relationship between antibody responses and severity. We have added additional data about disease severity to Table 1 and have included a new analysis (Table S2) in which we stratify the cohort based on asymptomatic versus mild symptom status. In the asymptomatic group, all significant differences between infants and mothers remained significant after stratification, but

significance was lost for comparisons of symptomatic infants versus mothers, likely because there are so few infants with symptoms (N = 3 or N = 2, depending on the specific experiment). We added text describing this in the Results section (lines 135-137, 176-177, and 247-248) and we now highlight this limitation and the need to extend this study to a cohort with a broader spectrum of disease severity in the Discussion section (line 334).

4. Based on the ADCC results, the authors claimed that “infants develop unique functional antibody repertoires during SARS-CoV-2 infection” (lines 239-240). At the same time, ADCC activity has a very high correlation with S-CEM cell surface staining (Figure S5F) but not neutralizing activity (Figure S5G). So it seems like the higher ADCC activity in infants is simply due to a higher level of non-neutralizing antibody. To claim that infants and mothers have difference “functional repertoires”, antibody level needs to be controlled for.

We thank the reviewer for raising this point and agree that “functional repertoires” implies that we have normalized the antibody responses with respect to total antibody level. To best represent the raw functional and binding activity in plasma in infants compared to mothers (in other words the concentration that the mother and infant have circulating, rather than the ratio), we have not normalized based on total antibody levels. We have changed the wording throughout the text to reflect that we are observing different levels of antibody binding and activity (that could reflect either differences in antibody concentration or potency), rather than a readout on the functional potency of the antibody repertoire. (See lines 34, 251 and 285).

Reviewer #2 (Remarks to the Author):

This is a well-written concise manuscript that compares antibody responses in mothers and their infants who acquired COVID-19 after delivery thus precluding any complication of placentally derived infant antibodies. They used the first positive sample in both infants and mothers. Results demonstrate differences in antibody responses including higher levels of Abs targeting the fusion domain of S2 and ADCC responses but more negative scaled differential selection scores indicating loss of Ab function against viral escape mutants in infants versus adults. The manuscript would be greatly strengthened if the authors included data on clinical disease in the cohort.

We thank the reviewer for positive and constructive feedback. We address specific concerns regarding clinical disease in the cohort below:

Specific questions:

1. Does disease severity correlate with Ab responses either in infants or mothers?

This is an important and interesting question raised by both reviewers (see also: point 2 from Reviewer 1). We have added additional data about disease severity and instances of mild symptoms in the cohort to Table 1 and have analyzed the impact of symptom status stratification on binding and ADCC responses in infants versus mothers (Table S2). In the asymptomatic group, all significant differences between infants and mothers remained significant after stratification, but significance was lost for comparisons of symptomatic infants versus mothers, likely because there are so few infants with symptoms (N = 3 or N = 2, depending on the specific experiment). Our cohort is further limited in that there were no severe cases. Given these features, we have insufficient power to answer this interesting question in this particular study, and we have added a sentence highlighting this limitation and the need for future study with a broader spectrum of disease severity in the Discussion (line 334).

2. Is there any information on viral shedding and does this contribute to differences in scaled differential selection. One might hypothesize that ongoing viral replication might select for viral escape mutants, which could impact Ab selection.

This is a very interesting point and we note there has been much debate about age-related differences in viral shedding during SARS-CoV-2 infection. In early large cohort studies there appeared to be no difference in viral shedding patterns, or slightly lower viral loads in children < 5 years of age, though these cohorts did not separate infants from older children (PMID 33949655, PMID 33542262). A study in Argentina including infants found significantly higher viral loads in infants versus older children or adults, suggesting unique viral dynamics in infants (PMID 34850028), though it is likely viral load is driven by disease severity in infants, as well (PMID 36071238). We don't have comprehensive information on viral loads in this cohort, though we now note this interesting hypothesis in the Discussion.

3. What is the mechanism by which age impacts antibody selection?

Mechanisms that influence age-related differences in antibody binding, activity, and selection are likely to be complex and could reflect major differences in immune developmental stage between infants and mothers, or possibly differences in pathogen exposure history and immune imprinting across age groups. Along these lines, elevated levels of ADCC activity in infants could be a result of either more abundant ADCC antibodies or higher potency ADCC antibodies. We have updated the discussion to include these possibilities as interesting avenues for further study, though we feel a complete mechanistic assessment is outside the scope of this cohort-based study.

4. Since this was part of a longitudinal cohort study, do the authors have subsequent samples to see if the immune response evolves over time?

Our group has previously published general antibody responses and their decay over time (Begnel et al.) In this previous study, Nucleocapsid antibody responses decayed rapidly, as has been described in many other studies, but there was not a major difference in mean time between participants' first positive plasma sample and loss of antibody detection among mothers and infants (9.7 months versus 8.1 months respectively). We have now added a short summary and reference to these data (lines 334-336).

REVIEWERS' COMMENTS

Reviewer #1 (Remarks to the Author):

The authors have addressed all my previous concerns.

Reviewer #2 (Remarks to the Author):

The authors have addressed all previous questions and limitations in the revised manuscript.